# GLUT3 Promotes Epithelial–Mesenchymal Transition via TGF-β/JNK/ATF2 Signaling Pathway in Colorectal Cancer Cells

**DOI:** 10.3390/biomedicines10081837

**Published:** 2022-07-29

**Authors:** Moon-Young Song, Da-Young Lee, Sun-Mi Yun, Eun-Hee Kim

**Affiliations:** College of Pharmacy and Institute of Pharmaceutical Sciences, CHA University, Seongnam 13488, Korea; wso219@naver.com (M.-Y.S.); angela8804@naver.com (D.-Y.L.); sun21mi@naver.com (S.-M.Y.)

**Keywords:** colorectal cancer, glucose transporter 3, epithelial–mesenchymal transition, c-Jun N-terminal kinase, activating transcription factor-2, transforming growth factor-β

## Abstract

Glucose transporter (GLUT) 3, a member of the GLUTs family, is involved in cellular glucose utilization and the first step in glycolysis. GLUT3 is highly expressed in colorectal cancer (CRC) and it leads to poor prognosis to CRC patient outcome. However, the molecular mechanisms of GLUT3 on the epithelial–mesenchymal transition (EMT) process in metastatic CRC is not yet clear. Here, we identified that activation of the c-Jun N-terminal kinase (JNK)/activating transcription factor-2 (ATF2) signaling pathway by transforming growth factor-β (TGF-β) promotes GLUT3-induced EMT in CRC cells. The regulation of GLUT3 expression was significantly associated with EMT-related markers such as E-cadherin, α- smooth muscle actin (α-SMA), plasminogen activator inhibitor-1 (PAI-1), vimentin and zinc finger E-box binding homeobox 1 (ZEB1). We also found that GLUT3 accelerated the invasive ability of CRC cells. Mechanistically, TGF-β induced the expression of GLUT3 through the phosphorylation of JNK/ATF2, one of the SMAD-independent pathways. TGF-β induced the expression of GLUT3 by increasing the phosphorylation of JNK, the nuclear translocation of the ATF2 transcription factor, and the binding of ATF2 to the promoter region of GLUT3, which increased EMT in CRC cells. Collectively, our results provide a new comprehensive mechanism that GLUT3 promotes EMT process through the TGF-β/JNK/ATF2 signaling pathway, which could be a potential target for the treatment of metastatic CRC.

## 1. Introduction

Colorectal cancer (CRC) is the third most common malignancy worldwide and it has been reported to be a major public health problem [1]. Approximately, 1.8 million new cases of CRC are initially diagnosed every year [2] and 5-year relative survival rate is the lowest in metastatic stage IV CRC at 15% [3]. Despite expanding clinical treatment and research effort, predictive biomarkers for CRC progression, metastasis and therapeutic strategy paradigms are not fully understood.

Epithelial–mesenchymal transition (EMT) is a process with biological relevance during which epithelial cells lose their characteristic cell–cell junctions to gain transitioning properties of mesenchymal cells. EMT is classified into three types: (1) embryonic development and organ formation [4]; (2) wound healing and fibrosis [5]; and (3) cancer progression [6]. EMT is a well-conserved developmental program, which has been involved in carcinogenesis and conferring metastatic ability to cancer cells by facilitating their mobility and invasion [7]. In cancer progression, EMT plays a crucial role in metastasis and is highly correlated with poor prognosis in patients in many types of cancers including CRC [8,9,10,11]. In addition, EMT-activated cancer cells can exhibit significant therapeutic resistance by acquiring stem cell competence [7]. Given these properties, the complex biological processes of EMT are thought to be a key feature of carcinogenesis, and targeting the EMT pathway could be an attractive strategy for the treatment of metastatic cancer.

Recent studies have revealed that metabolic reprogramming of cancer cells has emerged as a hallmark of cancer [12]. In cancer cells, it preferentially converts most glucose into lactate regardless of the presence of oxygen, a phenomenon referred to as the Warburg effect [12,13,14,15]. It differs distinctly from normal cells and requires large amounts of glucose as a metabolic fuel for enhanced glycolytic flux, mediating crucial bioactivities for their proliferation, survival, metastasis and malignant progression in cancer cells [15,16]. Glucose transporter (GLUT) mediates glucose entry by specific facilitative transporters into cells [17]. GLUT3 is well known as the one of 14 members of the SLC2 family of GLUTs, which exhibits a high glucose demand as well as the highest affinity for glucose uptake [18]. In fact, several studies have reported that the expression of GLUT3 is associated with cancer metastasis and poor prognosis [19]. Increased GLUT3 in high-grade gliomas has been reported to induce metastasis of human brain tumors [20]. In addition, it has been reported that high expression of GLUT3 triggers metastasis and may serve as a poor prognosis indicator in patients of non-small cell lung cancer (NSCLC) [21,22]. Moreover, several studies have reported a significant increase in the expression of GLUT3, promoting a marked enhancement in cancer metabolism and a poor survival rate in CRC patients [23,24,25]. However, the precise mechanisms responsible for the GLUT3-induced EMT and metastasis remain incompletely clarified.

Transforming growth factor-β (TGF-β) is a one of the major EMT-inducing factors that accelerate cancer progression, invasion and metastasis [26,27]. The ability of TGF-β-mediated EMT is generally initiated by ligand binding to two cognate heteromeric type II/type I receptor serine/threonine kinase complexes on the cancer plasma membrane [28]. These two receptors divide different types of signals into SMAD-dependent pathways and SMAD-independent pathways [29,30]. Several studies have shown that the TGF-β mediates the EMT process through the SMAD-dependent signaling pathway [31,32,33,34]. However, the evidence over the past few years have revealed that the TGF-β activates SMAD-independent pathways such as phosphatidylinositol 3′-kinase (PI3K)–protein kinase B (AKT), extracellular signal–regulated kinase 1 and 2 (ERK1/2), p38 mitogen-activated protein kinase (MAPK), and c-Jun N-terminal kinase (JNK) pathways [35,36,37,38]. JNK has been mainly studied for the function of apoptosis and anti-cancer effect under stress conditions [39]. Paradoxically, recent studies have shown that upregulation of JNK accelerates poor prognosis in cancer, which is associated with cell metastasis, invasion and the EMT process [40,41]. When JNK is stimulated by oncogenes and environmental stresses, it activates transcription factors such as p53, c-Jun, c-Myc and activating transcription factor-2 (ATF-2) [42]. ATF2 is known as a basic leucine zipper protein mediated by the JNK signaling pathway through phosphorylation of threonine residues within the NH_2_-terminal activation domain [43]. Additionally, ATF2 has been implicated in tumorigenesis, cell development, DNA damage response and the EMT process [44,45,46,47]. Although many studies have reported that JNK/ATF2 is involved in cancer metastasis and the EMT process, the molecular mechanism associated with GLUT3 has not yet been studied. In the present study, we report that GLUT3 promotes the EMT process through TGF-β/JNK/ATF2 signaling pathways, thereby accelerating the cancer progression and invasiveness in CRC cells.

## 2. Materials and Methods

### 2.1. Cell Culture

The human colorectal cancer cell lines (HCT116 and SW620) were obtained from the American Type Culture Collection (ATCC, Manassas, VA, USA) and Korean Cell Line Bank (KCLB, Seoul, Korea). HCT116 and SW620 cells were maintained according to the ATCC/KCLB’s instructions. Cells were maintained at 37 °C in a humidified atmosphere containing 5% CO_2_ and cultured in Roswell Park Memorial Institute (RPMI) 1640 Medium (GE Healthcare, Chicago, IL, USA) containing 10% (*v*/*v*) fetal bovine serum (ATCC), 100 U/mL penicillin and 100 μg/mL streptomycin.

### 2.2. Reagents

The JNK inhibitor, SP600125, was purchased from Cell Signaling Technology (Danvers, MA, USA). The TGF-β receptor kinase inhibitor, SB431542, was purchased from Tocris Cookson, Inc. (Ellisville, MO, USA). The antibodies specific to GLUT3, E-cadherin, N-cadherin, α-smooth muscle actin (α-SMA), connective tissue growth factor (CTGF), fibronectin, vimentin, zinc finger E-box binding homeobox 1 (ZEB1), β-actin, TGF-β RI, p-ATF2, ATF2, p-JNK and JNK were purchased from Santa Cruz Biotechnology (Dallas, TX, USA). The antibodies specific to plasminogen activator inhibitor-1 (PAI-1) were purchased from BD Bioscience (Franklin Lakes, NJ, USA). The antibodies specific to snail and twist were purchased from Abcam (Cambridge, UK). The antibodies specific to Nanog were purchased from Cell Signaling Technology (Danvers, MA, USA). The antibodies specific to OCT3/4 were purchased from R&D Systems (Minneapolis, MN, USA).

### 2.3. Lentiviral shRNA-Mediated Knockdown

SW620 cells were seeded in 100 mm dishes, and plasmid transfection was performed when the cells reached 70~80% confluence. Prior to transfection, SW620 cell culture medium was changed to antibiotic-free RPMI with 10% FBS. Ten micrograms of shSLC2A3 plasmid (Sigma, TRCN0000043615), 10 µg of Delta 320 packaging plasmid, and 5 µg of SV g Lenti-virus envelope plasmid were cotransfected using 20 µL of Lipofectamine 2000 transfection reagent (Invitrogen, Waltham, MA, USA). The next day after transfection, the transfection medium was changed to fresh complete-culture medium. The virus supernatant was collected every 18 h, filtered through a 0.45 mm filter, and frozen at −80 °C until further usage. For lentiviral transfection, 1 × 10^6^ cells were seeded in 100 mm dishes and incubated with the virus at a multiplicity of infection (MOI) of ~1 for 18 h in the presence of 8 µg/mL polybrene (Santa Cruz, Dallas, TX, USA). Seventy-two hours later, the cells were selected with 2.5 µg/mL puromycin (Invitrogen, Waltham, MA, USA) for 3 d. The cells after puromycin selection were used in downstream experiments. The sequences of three SLC2A3 targeting shRNA were as follows: SLC2A3 shRNA, TRCN0000043615, 5′-CCG GAG TAG CTA AGT CGG TTG AAA TCT CGA GAT TTC AAC CGA CTT AGC TAC TTT TTT G-3′.

### 2.4. Quantitative Reverse Transcriptase Polymerase Chain Reaction (qRT-PCR) Analysis

Total mRNA was extracted from cells using TRIzol^TM^ Reagent (Thermo Fisher Scientific, Waltham, MA, USA), according to the manufacturer’s protocol. Reverse transcription was performed with 2 μg of pure RNA using Labopass cDNA synthesis kit (Cosmogenetech, Seoul, Korea). qRT-PCR was performed on a ViiATM 7 Real-time PCR system (Applied Biosystems, Waltham, MA, USA) using Luna universal qPCR master mix (New England Biolabs, Beverly, MA, USA). The relative quantities of target genes were calculated from triplicate samples after normalization to an internal control, 18S ribosomal RNA (18S rRNA). All oligonucleotide primers, listed in Table 1 below, were purchased from Macrogen company.

### 2.5. Western Blot Analysis

Whole cells were washed twice in cold PBS and lysed in cell lysis buffer (Cell Signaling, Danvers, MA, USA) plus phosphatase and protease inhibitors (Roche Applied Science, Mannheim, Germany). The cells were vortexed and incubated on ice for 10 min and centrifuged for 15 min at 13,000 rpm. Supernatants were collected and stored at −80 °C. The protein concentration of each supernatant was quantified using protein assay reagent from the BCA assay (Thermo Fisher Scientific, Waltham, MA, USA) in accordance with the manufacturer’s instructions. Eluted proteins were separated by SDS–polyacrylamide gel electrophoresis (10%) and transferred to polyvinylidene fluoride membranes. Membranes (PVDF) were blocked with PBS containing 0.1% Tween (PBST) and 3% bovine serum albumin for 1 h and then with primary antibodies diluted in the same buffer and incubated with the primary antibodies overnight at 4 °C. Next day, the membrane was washed with PBST, incubated with peroxidase-conjugated secondary antibodies, rewashed, and then visualized using an enhanced chemiluminescence system (Thermo Fisher Scientific, Waltham, MA, USA) and LAS-4000 imager (GE Healthcare Life Sciences, Piscataway, NJ, USA).

### 2.6. Plasmids Cloning

Primers for human GLUT3 plasmid cloning were as follows: GLUT3-forward, 5′-ATG CAA GCT TAT GGG GAC ACA GAA GGT CAC-3′; GLUT3-reverse, 5′-ATG CGG ATC CCG ACA TTG GTG GTG GTC TCC T-3′. DNA extracted from SW620 human CRC cell line was used as the template for PCR and amplified using PrimeSTAR (Takara Bio Inc., Shiga, Japan). The GLUT3 PCR fragment was digested with HindIII and BamHI and then cloned into pEJ-3HA vector (pEGFP origin, Invitrogen, Waltham, MA, USA). Restriction enzymes were purchased from NEB (Ipswich, MA, USA). The resulting GLUT3/pEJ-3HA were confirmed by sequencing. Primers for human GLUT3 promoter cloning were as follows: GLUT3pro-forward, 5′-ATG CGG TAC CCC GAT TAT CCC TCC CTC AGT-3′; GLUT3pro-reverse, 5′-ATG CCT CGA GTA TCA GAG CTG GGG TGA CCT-3′. Genomic DNA extracted from normal human fibroblasts was used as the template for PCR. Promoters were amplified using PrimeSTAR (Takara Bio Inc., Shiga, Japan). The PCR fragment of GLUT3 was digested with KpnI and XhoI and then cloned into pGL3-basic vector (Promega, Madison, WI, USA). Restriction enzymes were purchased from NEB (Ipswich, MA, USA). The resulting pGL3-GLUT3 vectors were confirmed by sequencing.

### 2.7. Transient Transfection and the Luciferase Reporter Assay

Human colon cancer cells were transiently transfected with GLUT3/pEJ-3HA plasmid, E-cadherin/pGL3-Luc, α-SMA/pGL3-Luc, GLUT3/pGL3 and ATF/cAMP-response element (CRE) mutant GLUT3/pGL3-Luc plasmid using Lipofectamine^®^ 2000 Transfection Reagent (Invitrogen, Waltham, MA, USA) following the manufacturer’s instructions. The ATF/CRE binding sites (−716/−708) were mutated with GLUT3 mutant-forward, 5′-GAG CCG AGA TCG CAC CAA TAA AAT CCA GCC TGG GCG ACA G-3′; GLUT3 mutant-reverse, 5′-CTG TCG CCC AGG CTG GAT TTT ATT GGT GCG ATC TCG GCT C-3 using site-directed mutagenesis. Twenty-four hours after transfection, cells were collected and assayed for luciferase activity using the luciferase assay system (Promega, Madison, WI, USA) according to the manufacturer’s instructions. Each experiment was repeated in triplicate.

### 2.8. Invasion Assay

The invasion assay was conducted using the Transwell system (Corning Inc., Corning, NY, USA). The inside of the transwell plates were coated with 0.1% gelatin (Sigma-Aldrich, St. Louis, MO, USA). Cells were harvested and seeded at a density of 1 × 10^4^ cells/well in 0.2 mL of serum-free medium in the transwell top chamber, and the lower chamber was filled with 0.5 mL of medium containing 10% FBS. After incubation for 48 h, the lower membrane was fixed with 95% ethanol, stained with 0.2% crystal violet (Sigma-Aldrich, St. Louis, MO, USA) for 30 min at a room temperature, washed with PBS and observed using a microscope. Data were recorded from three random fields of the lower membrane surface and were analyzed in triplicate.

### 2.9. Chromatin Immunoprecipitation Assay

Cells (approximately 1 × 10^7^) were cross-linked with formaldehyde solution, collected in PBS, resuspended in lysis buffer and sonicated on ice. The lysates were then diluted with chromatin immunoprecipitation (ChIP) dilution buffer, pre-cleared with protein A agarose, and then incubated with ATF2 antibodies (Santa Cruz, Dallas, TX, USA) overnight. The immune complexes were collected with protein A agarose, cross-links were reversed and DNA was recovered. Relative amounts of DNA in the complex were quantified by RT-PCR. The PCR was performed using the following primers below: GLUT3 ChIP-forwards, 5′-GGC GCC TAT AGT CCC AGC TAC TCG-3′; GLUT3 ChIP-reverse, 5′-ATC AGG GCA GTA GGA ATA AGA GGT-3′.

### 2.10. Statistical Analysis

Results are expressed as the mean ± standard deviation (SD). The statistical significance was analyzed using one-way analysis of variance (ANOVA). Statistical significance was accepted at *p* < 0.05.

## 3. Results

### 3.1. GLUT3 Regulates the Expression of EMT-Related Genes and Promotes Invasiveness in CRC Cells

To investigate whether GLUT3 is involved in the EMT, invasiveness and metastasis of CRC cells, HCT116 cells were stably transduced with a vector encoding GLUT3 and an empty vector. The enforced expression of GLUT3 induced a shift in epithelial to mesenchymal markers with a loss of cell-to-cell-contact-related genes. As shown in Figure 1A, E-cadherin was significantly decreased in GLUT3-overexpressed cells, whereas EMT-related markers such as N-cadherin, snail, twist, α-SMA, CTGF, fibronectin 1, vimentin, PAI-1 and ZEB-1 were significantly increased. In parallel with protein levels, the mRNA transcript of mesenchymal markers including snail, twist, α-SMA, CTGF, fibronectin, vimentin, PAI-1 and ZEB1 were dramatically increased by transfection with GLUT3 in CRC cells. However, the mRNA expression of N-cadherin was not significantly changed, whereas the epithelial marker E-cadherin was significantly decreased by transfection with GLUT3 in HCT116 cells (Figure 1B). Next, we examined the effect of GLUT3 on the promoter activity of E-cadherin and α-SMA. Consequently, the upregulation of GLUT3 levels inhibited the promoter activity of E-cadherin and promoted the promoter activity of α-SMA (Figure 1C). To further explore the activity of GLUT3 in the regulation of metastasis, we performed a transwell invasion assay in HCT116 cells transfected with GLUT3. As shown in Figure 1D, we observed that the invasive potential of HCT116 cells was significantly increased in GLUT3-transfected CRC cells.

To further examine whether the loss of endogenous GLUT3 could suppress EMT-related markers, we established shRNA-mediated GLUT3 knockdown SW620 cells. The decreased expression of GLUT3 strongly reduced the expression of mesenchymal markers N-cadherin, snail, twist, α-SMA, CTGF, fibronectin 1, vimentin, PAI-1 and ZEB-1, whereas it induced the expression of the epithelial marker E-cadherin (Figure 2A). Furthermore, the expression of EMT-related genes such as N-cadherin, snail, twist, α-SMA, CTGF, fibronectin, vimentin, PAI-1 and ZEB1 was decreased at the mRNA levels and the mRNA expression of E-cadherin was significantly increased in shRNA-mediated GLUT3 knockdown SW620 cells (Figure 2B). To examine the role of GLUT3 on cell mobility, we next performed the transwell invasion assay in GLUT3-knockdown SW620 cells. As a result, the knockdown of GLUT3 significantly reduced the invasive ability of SW620 cells (Figure 2C). These findings suggest that GLUT3 plays an important role in mediating the invasion and EMT process through regulation of the expression of EMT-related genes in CRC cells.

### 3.2. GLUT3 Induces the Expression of Stemness Makers in CRC Cells

Multiple lines of evidence support that elevated EMT may confer stem cell properties in cancer cells [7]. GLUT3-induced expression of EMT-related genes and invasion of CRC cells prompted us to determine whether GLUT3 is associated with the acquisition of stem cell properties in CRC cells. We investigated the effect of GLUT3 on the expression of stemness markers in GLUT3-overexpressed CRC cells. As a result, the stable overexpression of GLUT3 strongly increased the protein expression of master regulators of the stemness markers, Nanog and OCT3/4 (Figure 3A). In parallel with the elevated expression of proteins, the mRNA transcript of stemness markers was also increased by the transfection with GLUT3 in CRC cells (Figure 3B). Upon knockdown of GLUT3 expression by employing the GLUT3 shRNA, however, the protein levels of the stemness markers Nanog and OCT3/4 were significantly decreased (Figure 3C). As shown in Figure 3D, transfection with GLUT3 shRNA abolished the mRNA expression of Bmi-1 and OCT3/4 in SW620 cells. These results indicate that GLUT3 contributes to the regulation of the expression of cancer stemness markers.

### 3.3. TGF-β Is the Upstream Regulator of GLUT3-Induced EMT in CRC Cells

To investigate the molecular mechanism of whether TGF-β regulates GLUT3-induced EMT, HCT116 cells were exposed to SB431542, a TGF-β receptor 1 antagonist. We then examined the protein expression of GLUT3 and EMT-related molecules. As shown in Figure 4A, the treatment of HCT116 cells with SB431542 significantly reduced the expression of TGF-β R1 and GLUT3. The expression of EMT-related markers such as α-SMA, ZEB1 and PAI-1 were also diminished in HCT116 cells treated with SB431542. Next, we determined the effect of SB431542 on the invasive potential of CRC cells. The GLUT3-induced enhancement in invasion was markedly reduced by the treatment with TGF-β R1 inhibitor SB431542 (Figure 4B). Taken together, these results suggest that activation of the TGF-β signaling pathway accelerates the invasive ability of CRC cells through the upregulation of GLUT3 and EMT-related factors.

### 3.4. TGF-β Regulates GLUT3-Induced EMT through JNK/ATF2 Signaling Pathway in CRC Cells

To determine the precise molecular mechanism and the link between TGF-β and GLUT3 in colorectal cancer, we investigated the downstream signalings of the TGF-β pathway, JNK and the transcription factor ATF2. To investigate whether the JNK/ATF2 pathway is the downstream of TGF-β, TGF-β inhibitor SB431542 was utilized in HCT116 cells. As shown in Figure 5A, the phosphorylation of JNK and ATF2 was attenuated by the treatment with TGF-β inhibitor SB431542. Therefore, we wondered whether GLUT3 was induced by the treatment with exogenous TGF-β, and whether TGF-β was the truthful upstream regulator of the expression of GLUT3. As a result, the treatment of HCT116 cells with TGF-β significantly induced the expression of GLUT3 mRNA in CRC cells (Figure 5B).

Next, we identified the exact mechanism by which ATF2 regulates the expression of GLUT3 under TGF-β activation and searched the promoter region of the human GLUT3 gene to find a consensus for the ATF/CRE region. To confirm the effect of TGF-β on GLUT3 promoter activity, we performed a luciferase reporter gene assay. The cells stimulated with TGF-β exhibited significantly enhanced ATF/CRE luciferase activity (Figure 5C). However, mutation of the ATF/CRE core sequence, which lacks the DNA-binding region, did not profoundly change the luciferase reporter activity even in the treatment with TGF-β (Figure 5C). These results indicate that the ATF/CRE sequence in the GLUT3 promoter region may be responsible for ATF2 binding. To further assess the role of the transcription factor ATF2 in TGF-β-induced GLUT3 expression, we performed a ChIP assay to determine the binding ATF2 to the GLUT3 promoter region. As shown in Figure 5D, the direct binding of ATF2 to the ATF/CRE consensus region of the GLUT3 promoter was observed. Our findings indicate that ATF2 drives the transcription of GLUT3 through direct binding to the promoter region of GLUT3 after TGF-β activation. Taken together, these results suggest that ATF2 plays a crucial role in mediating the TGF-β-induced expression of GLUT3.

To define the regulation of the TGF-β/JNK/ATF2 axis in GLUT3-induced EMT, we explored the role of JNK. For this purpose, we utilized SP600125, a chemical inhibitor of JNK (Figure 6A). Treatment of HCT116 cells with 10 μM of SP600125 significantly attenuated the phosphorylation of ATF2 and also the expression of GLUT3 (Figure 6A). To further verify the role of JNK in the induction of invasion, we treated GLUT3-overexpressed HCT116 cells with the JNK inhibitor SP600125. We conducted an invasion analysis to evaluate the role of JNK in the metastatic activity of GLUT3 in CRC cells. As illustrated in Figure 6B, the GLUT3-induced enhancement in invasion was markedly reduced by SP600125. These results demonstrate that JNK plays an important role in mediating TGF-β-induced EMT and invasion through the regulation of transcriptional activity of ATF2.

## 4. Discussion

Over the past few decades, various studies of cancer metabolism have been devoted to understanding and predicting prognosis related to growth factors. Among the factors related to cancer metabolism, GLUT3 has been well-known to control the uptake of glucose and glycolysis in various cancer cells [19,20,21,22,23,24,25]. The upregulation of GLUT3 has been reported to be associated with poor prognosis in many human types of cancer, including CRC [19,48,49,50]. Recent studies have revealed that GLUT3 is strongly upregulated during the EMT process, which plays an important role in the poor prognosis in cancer [24,49,50]. Jiang et al. reported that GLUT3 may regulate the EMT process through the yes-associated protein (YAP), as a major downstream effector in the Hippo pathway and the AMPK pathway in human CRC tissues and CRC cells [24]. In addition, Kuo et al. reported that the activation of the GLUT3-YAP pathway reprograms cancer metabolism acting as a master stimulator, thereby promoting metastasis [50]. There is also a report that GLUT3, but not GLUT1, correlates with poor survival in brain tumors [48]. However, the mechanism for the EMT induced by GLUT3 overexpression is still unclear. Therefore, detailed molecular-mechanism studies on the relationship between GLUT3 and EMT process are needed. In this report, we focused on the molecular mechanisms of GLUT3 for the EMT process in metastatic CRC through the JNK/ATF2 signaling pathway mediated by TGF-β activation. We also suggest that the TGF-β/JNK/ATF2 signaling pathway is an important therapeutic target for GLUT3-induced metastasis in CRC cells.

TGF-β, as a potent inducer of EMT, abrogates cell–cell adhesion and accelerates the migration and invasion of cancer cells [51]. According to revealed studies, the activation of TGF-β promotes the EMT response, which is not only the downregulation of epithelial markers such as E-cadherin, but also the upregulation of mesenchymal markers including N-cadherin, α-SMA, and vimentin following tumor progression [52,53]. Our study showed that the expression of GLUT3 significantly contributes to the EMT process by promoting mesenchymal characteristics and triggering cancer cell invasion in CRC cells. Interestingly, we found that inhibition of TGF-β reduced not only the expression of GLUT3 but also the expression of EMT-related markers such as α-SMA, ZEB1 and PAI-1 as well as the potential invasive ability of genetically GLUT3-overexpressed CRC cells. These findings suggest that the activation of the TGF-β signaling pathway promotes the invasive potential of CRC cells through the upregulation of GLUT3 and EMT-related factors. These results are in agreement with the previous studies demonstrating that TGF-β pathway regulates the EMT process through GLUT3 activation in cancer [54].

Multiple lines of evidence indicate that the TGF-β-induced EMT is mediated by SMAD-dependent pathway or by SMAD-independent pathways in a variety of systems [29,30,35,55]. Zhang et al. revealed that the upregulation of TGF-β promotes gastric cancer metastasis through SMAD2/3 signaling, which enhances migration, invasion and the EMT process in gastric cancer cells [31]. Yu et al. showed that TGF-β-induced EMT, invasiveness and metastasis were prevented by SMAD7, known as a negative regulator of TGF-β signaling in metastatic cancer cells [32]. In hepatocellular carcinoma, Qu et al. demonstrated the invasion and metastasis of hepatocellular carcinoma against the EMT process via the TGF-β/SMAD signaling pathway in vitro and in vivo [33]. In addition, Yeh et al. demonstrated that TGF-β and SMAD2/3 pathway is involved in the promotion of EMT, stemness and metastasis in liver cancer cells [34]. These results show that TGF-β-induced EMT is regulated by a SMAD-dependent pathway in various cancer cells.

TGF-β, meanwhile, may also regulate the EMT process through SMAD-independent pathways such as MAPKs, namely ERK1/2, p38, and the JNK signaling pathway [30,56]. Recent studies revealed that TGF-β accelerated the invasive and migratory potential through the induction of EMT via the ERK pathway in breast cancer cells [57] and glioma cells [58]. In addition, TGF-β induced the EMT process through the p38 pathway in metastatic lung cancer [59,60] and cervical cancer cells [61]. Moreover, JNK signaling is also associated with TGF-β-induced EMT in various cancer cells [62,63]. Furthermore, the phosphorylation of JNK has been reported to be regulated by TGF-β activation, and the EMT process is promoted after the activation of JNK in response to TGF-β [64,65,66,67,68]. In the present study, TGF-β inhibition abrogated the phosphorylation of JNK and the invasive property of GLUT3-overexpressed HCT116 cells, suggesting that TGF-β is an upstream regulator of the JNK pathway. These findings demonstrate that TGF-β regulates the EMT process through JNK activation and the subsequent GLUT3 expression, a SMAD-independent pathway in human CRC cells.

After establishing the GLUT3-induced EMT process by the activation of the TGF-β/JNK signaling pathway, we explored which transcription factors could lead to the upregulation of GLUT3 by the TGF-β/JNK signaling pathway. Previous studies have shown that transcription factors belonging to ATF2 and c-jun/c-fos were implicated in TGF-β regulation [68,69]. JNK induces the phosphorylation of ATF2 and c-jun, a component of transcription factors correlated with TGF-β, and plays an important role in the EMT process [64,70,71,72]. As a target of the JNK signaling pathway, ATF2 binds to the promoter region of various genes, which can be phosphorylated by JNK activity on two threonine residues in the NH2-terminal activation domain [43]. Additionally, several studies established that TGF-β activation accelerates ATF2 in various types of cancers. Our demonstration of the activation of ATF2 mediated by TGF-β in human colorectal cancer (HCT116) cells is in agreement with several previous studies, implying the metastatic potential of elevated ATF2. According to Xu et al. [73], ATF2 promoted the EMT process by treatment with the TGF-β inducer in pancreatic cancer cells. Similarly, the activation of ATF2 predicted poor prognosis and promoted the malignant phenotypes in renal cancer cells [47]. In this study, we identified ATF2 as a transcription factor for the expression of GLUT3, mediated by the phosphorylation of JNK upon the activation of TGF-β. Our findings demonstrate that ATF2 directly binds to the ATF/CRE consensus region of the GLUT3 promoter, thereby inducing the expression of GLUT3 levels; however, ATF2 did not affect the expression of GLUT3 in HCT116 cells transfected with constructs with the mutated ATF/CRE binding site. These results suggest that TGF-β triggers the phosphorylation of JNK, leading to the transcriptional activation of ATF2, which may contribute to the EMT process and metastatic potential through increased GLUT3 expression in colorectal cancer.

Collectively, our results show that TGF-β positively regulates JNK, which promotes ATF2 to induce the transcription of GLUT3, and consequently, this increase in GLUT3 expression exacerbates the EMT in CRC cells. These findings suggest that novel molecular mechanisms of GLUT3 for the EMT process act via the TGF-β/JNK/ATF2 pathway, thereby exacerbating the invasive ability of cancer cells in metastatic colorectal cancer (Figure 7).

## 5. Conclusions

In conclusion, we observed the correlation of GLUT3 and the EMT process in human CRC cells in this study. The elevated levels of GLUT3 during cancer metabolic reprogramming are likely to provoke the EMT process and stemness acquisition, thereby upregulating invasiveness and metastasis in CRC cells. We show here a detailed molecular mechanism by which GLUT3 is regulated by the activation of the JNK/ATF2 pathway during TGF-β-induced EMT. Taken together, these findings provide a novel mechanism underlying the EMT induction and metastatic potential of GLUT3 that links the TGF-β-JNK-ATF2 axis.

## Figures and Tables

**Figure 1 biomedicines-10-01837-f001:**
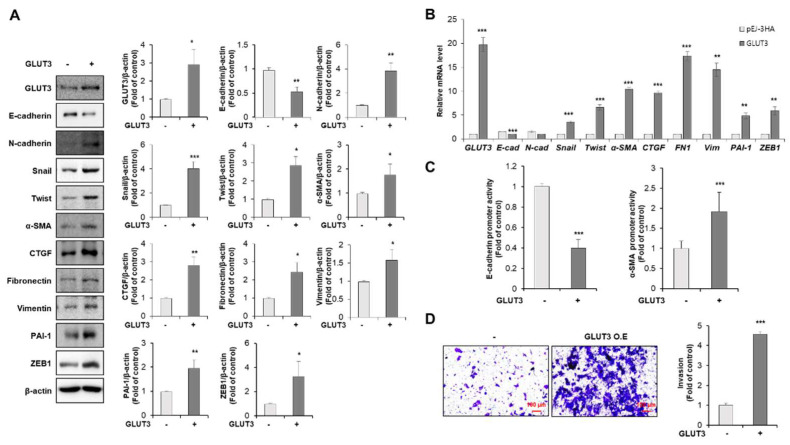
GLUT3 promotes EMT in HCT116 cells. (**A**) HCT116 cells were transfected with GLUT3/pEJ-3HA (2 µg) for 24 h. Western blot analysis was performed to measure the protein levels of GLUT3, E-cadherin, N-cadherin, snail, twist, α-SMA, CTGF, fibronectin, vimentin, PAI-1 and ZEB1. (**B**) qRT-PCR was performed to measure the mRNA levels of GLUT3, E-cadherin (E-cad), N-cadherin (N-Cad), snail, twist, α-SMA, CTGF, fibronectin 1 (FN1) and vimentin (Vim), PAI-1 and ZEB1. (**C**) The E-cad-Luc or α-SMA-Luc plasmids were transfected with GLUT3/pEJ-3HA into HCT116 cells for 24 h. Cell extracts were harvested, and the luciferase assay was performed. (**D**) Cell invasion was analyzed after 48 h post transfection with GLUT3/pEJ-3HA (2 µg) to allow for the permeabilization of the transwell membrane. The membrane was stained with 0.2% crystal violet. Scale bar, 100 µm. GLUT3, glucose transporter 3; α-SMA, α-smooth muscle actin; PAI-1, plasminogen activator inhibitor-1; ZEB1, zinc finger E-box binding homeobox 1, qRT-PCR, quantitative reverse transcription PCR; CTGF, connective tissue growth factor. Data are the mean ± standard deviation. Statistical significance was analyzed by analysis of variance. * *p* < 0.05, ** *p* < 0.01, and *** *p* < 0.001, significantly different compared with control.

**Figure 2 biomedicines-10-01837-f002:**
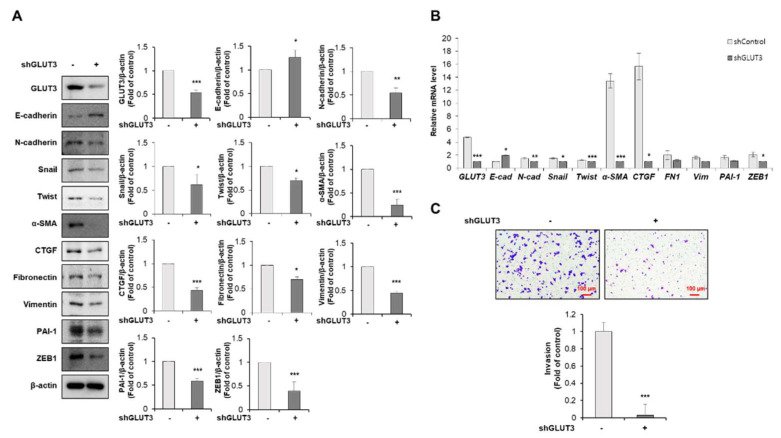
Knockdown of GLUT3 in SW620 cells reduces EMT process. (**A**) Western blot analysis was conducted for measuring the protein levels of GLUT3, E-cadherin, N-cadherin, snail, twist, α-SMA, CTGF, fibronectin 1 and vimentin, PAI-1 and ZEB1 in shGLUT3-induced SW620 cells. (**B**) qRT-PCR was conducted to measure the mRNA levels of GLUT3, E-cadherin (E-cad), N-cadherin (N-Cad), snail, twist, α-SMA, CTGF, fibronectin 1 (FN1) and vimentin (Vim), PAI-1 and ZEB1 in shGLUT3-transfected SW620 cells. (**C**) shGLUT3 was analyzed after 48 h using cell invasion assay to allow for the permeabilization of the transwell membrane. The membrane was stained with 0.2% crystal violet. Scale bar, 100 µm. GLUT3, glucose transporter 3; EMT, epithelial–mesenchymal transition; α-SMA, α-smooth muscle actin; PAI-1, plasminogen activator inhibitor-1; ZEB1, zinc finger E-box binding homeobox 1, qRT-PCR, quantitative reverse transcription PCR; CTGF, connective tissue growth factor. Data are the mean ± standard deviation. Statistical significance was analyzed by analysis of variance. * *p* < 0.05, ** *p* < 0.01, and *** *p* < 0.001, significantly different compared with control.

**Figure 3 biomedicines-10-01837-f003:**
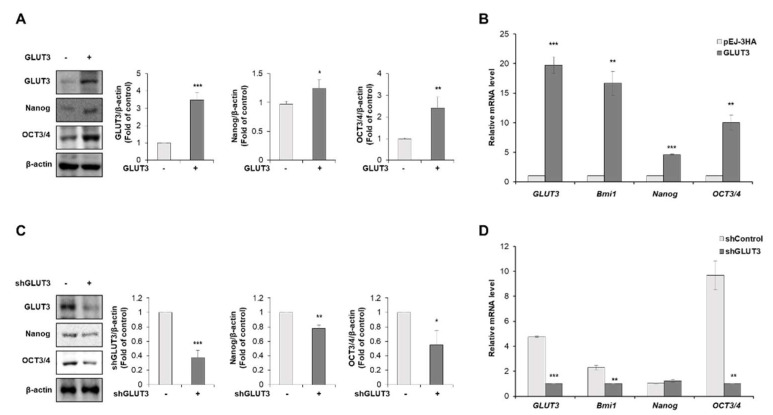
Overexpression and knockdown of GLUT3 regulates stemness markers in CRC cells. (**A**) HCT116 cells were transfected with GLUT3/pEJ-3HA (2 µg) for 24 h. Western blot analysis was conducted for measuring the protein levels of GLUT3, Nanog and OCT3/4. (**B**) qRT-PCR was conducted to measure the mRNA levels of GLUT3, Bmi1, Nanog and OCT3/4. (**C**) Western blot analysis was conducted for measuring the protein levels of GLUT3, Nanog and OCT3/4 in shGLUT3-transfected SW620 cells. (**D**) qRT-PCR was conducted to measure the mRNA levels of GLUT3, Bmi1, Nanog and OCT3/4 in shGLUT3-transfected SW620 cells. GLUT3, glucose transporter 3; CRC, colorectal cancer; qRT-PCR, quantitative reverse transcription PCR. Data are the mean ± standard deviation. Statistical significance was analyzed by analysis of variance. * *p* < 0.05, ** *p* < 0.01 and *** *p* < 0.001, significantly different compared with control.

**Figure 4 biomedicines-10-01837-f004:**
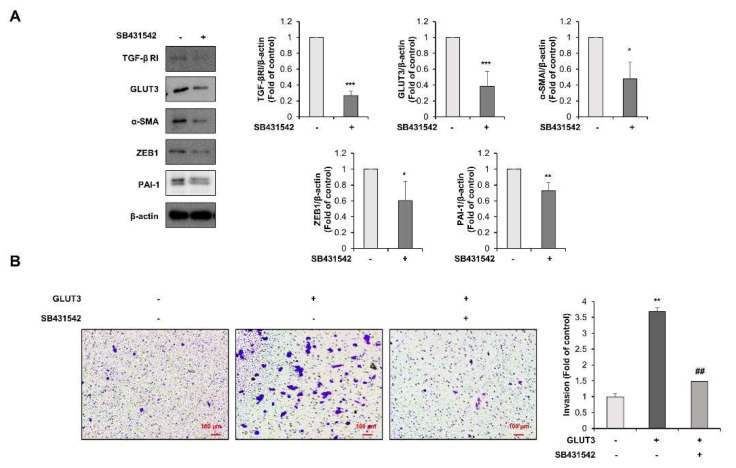
TGF-β signaling pathway regulates GLUT3-induced EMT in CRC cells. (**A**) Lysates of the HCT116 cells treated or not with 10 μM SB431542 for 48 h. Western blot analysis was conducted for measuring the protein levels of TGF-β RI, GLUT3, α-SMA, ZEB1 and PAI-1. (**B**) Cell invasion was analyzed after 24 h post transfection with GLUT3/pEJ-3HA (1.5 µg) and treated or not with 10 μM SB431542 for 24 h to allow for the permeabilization of the transwell membrane. The membrane was stained with 0.2% crystal violet. Scale bar, 100 µm. TGF-β, transforming growth factor-β; GLUT3, glucose transporter 3; EMT, epithelial–mesenchymal transition; TGF-β RI, transforming growth factor-β receptor I; α-SMA, α-smooth muscle actin; ZEB1, zinc finger E-box binding homeobox 1, PAI-1, plasminogen activator inhibitor-1. Data are the mean ± standard deviation. Statistical significance was analyzed by analysis of variance. * *p* < 0.05, ** *p* < 0.01, and *** *p* < 0.001, significantly different compared with control; ## *p* < 0.01 compared to the GLUT3 group.

**Figure 5 biomedicines-10-01837-f005:**
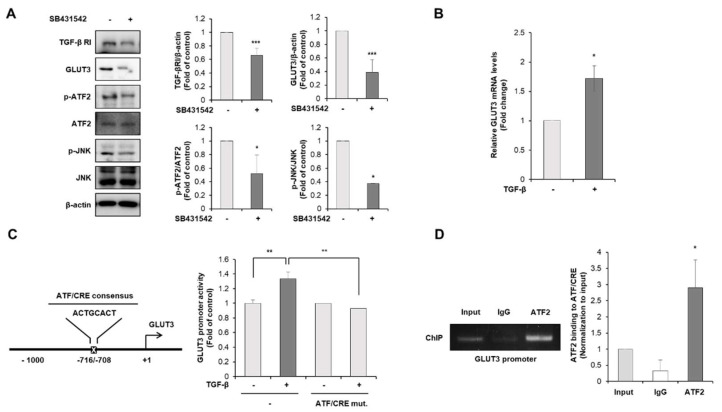
ATF2 directly regulates GLUT3 transcription through TGF-β activation. (**A**) Lysates of the HCT116 cells treated or not with 10 μM SB505124 for 48 h. Western blot analysis was conducted for measuring the protein levels of TGF-β RI, GLUT3, p-ATF2, ATF2, p-JNK and JNK. (**B**) qRT-PCR was conducted to measure the mRNA levels of GLUT3 after being treated or not with TGF-β (10 ng/mL) for 24 h. (**C**) HCT116 cells were transiently transfected with GLUT3/pGL3 and ATF/CRE mutant GLUT3/pGL3 luciferase promoter plasmids for 24 h and treated or not with TGF-β (10 ng/mL) for 24 h. Cell extracts were harvested, and the luciferase assay was performed. (**D**) Effect of TGF-β activation on ATF2 binding to the GLUT3 chromatin. Cross-linked chromatin was immunoprecipitated with antibodies against ATF2 or rabbit lgG and analyzed by RT-PCR using primers that flank the ATF/CRE binding site. ATF2, activating transcription factor-2; p-ATF2, phospho-ATF2; GLUT3, glucose transporter 3; TGF-β, transforming growth factor-β; TGF-β RI, transforming growth factor-β receptor I; JNK, c-Jun N-terminal kinase; p-JNK, phospho-JNK; CRE, cAMP-response element; qRT-PCR, quantitative reverse transcription PCR; IgG, immunoglobulin G. Data are the mean ± standard deviation. Statistical significance was analyzed by analysis of variance. * *p* < 0.05, ** *p* < 0.01, and *** *p* < 0.001, significantly different compared with control.

**Figure 6 biomedicines-10-01837-f006:**
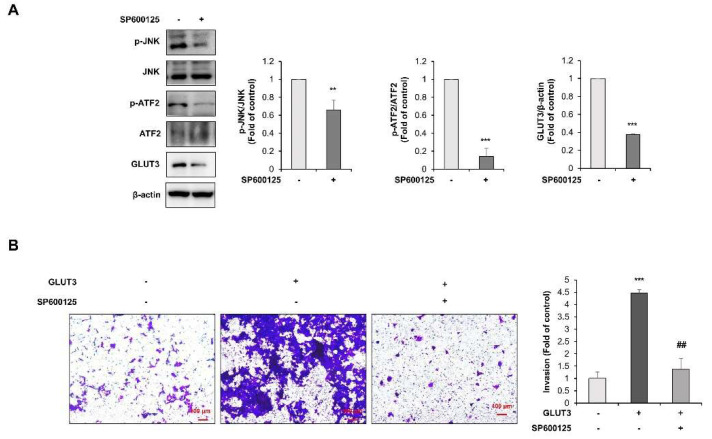
JNK is an upstream kinase regulating ATF2 in GLUT3-induced EMT in CRC cells. (**A**) Lysates of the HCT116 cells treated or not with SP600125 (20 μM) for 24 h. Western blot analysis was conducted for measuring the protein levels of p-JNK, JNK, p-ATF2, ATF2 and GLUT3. (**B**) Cell invasion was analyzed after 24 h post transfection with GLUT3/pEJ-3HA (2 µg) and treated or not with SP600125 (20 μM) for 24 h to allow for the permeabilization of the transwell membrane. The membrane was stained with 0.2% crystal violet. Scale bar, 100 µm. JNK, c-Jun N-terminal kinase; p-JNK, phospho-JNK; ATF2, activating transcription factor-2; p-ATF2, phospho-ATF2; GLUT3, glucose transporter 3; CRC, colorectal cancer. Data are the mean ± standard deviation. Statistical significance was analyzed by analysis of variance. ** *p* < 0.01 and *** *p* < 0.001, significantly different compared with control; ## *p* < 0.01 compared to GLUT3 group.

**Figure 7 biomedicines-10-01837-f007:**
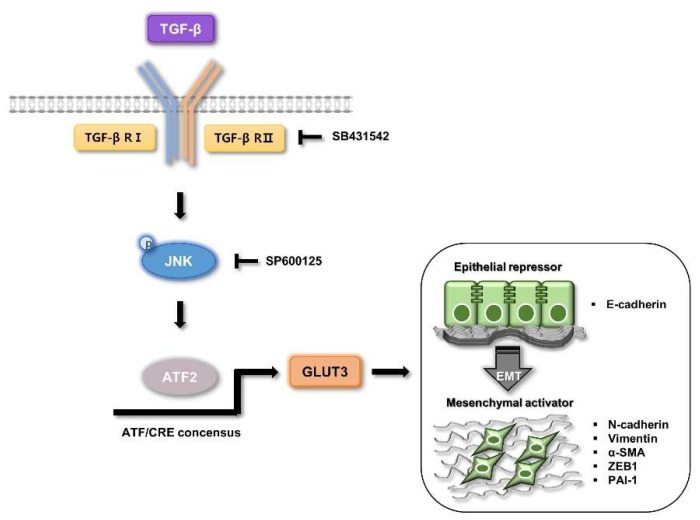
Schematic representation of the regulation of GLUT3 on TGF-β-mediated EMT in CRC cells. Expression of GLUT3 is regulated by the TGF-β/JNK/ATF2 signaling pathways, and overexpression of GLUT3, thereby, exacerbates CRC cells’ invasiveness. “↓” indicates induction; “├” indicates inhibition. GLUT3, glucose transporter 3; TGF-β, transforming growth factor-β; EMT, epithelial–mesenchymal transition; CRC, colorectal cancer; JNK, c-Jun N-terminal kinase; ATF2, activating transcription factor-2.

**Table 1 biomedicines-10-01837-t001:** Primer sequences used for qRT-PCR.

Species	Gene	Primer Sequence	
Human (qRT-PCR)	18S rRNA	Forward	GCAATTATTCCCCATGAACG
		Reverse	GGCCTCACTAAACCATCCAA
	GLUT3	Forward	TTGCTCTTCCCCTCCGCTGC
		Reverse	ACCGTGTGCCTGCCCTTCAA
	CDH1 (E-cadherin)	Forward	TCC CCG GCC AGC CAT
		Reverse	GCA GAG CCA AGA GGA GAC C
	CDH2 (N-cadherin)	Forward	GAG GCT TCT GGT GAA ATC GC
		Reverse	AGA AGA GGC TGT CCT TCA TGC
	Snail	Forward	GCTGCAGGACTCTAATCCAGA
		Reverse	ATCTCCGGAGGTGGGATG
	Twist	Forward	GGCATCACTATGGACTTTCTCTATT
		Reverse	GGCCAGTTTGATCCCAGTATT
	α-SMA	Forward	CAGTGGAATGCAGTGGAAGA
		Reverse	AGGGAAGCTGAAAGCTGAAG
	CTGF	Forward	AGG ATG TGC ATT CTC CAG CC
		Reverse	GCC ACA AGC TGT CCA GTC TA
	Fibronectin1	Forward	GAACTATGATGCCGACCAGAA
		Reverse	GGTTGTGCAGATTTCCTCGT
	Vimentin	Forward	GCT TCA GAG AGA GGA AGC CG
		Reverse	AAG GTC AAG ACG TGC CAG AG
	Bmi1	Forward	TGAAGATAGAGGAGAGGTTGC
		Reverse	CTGCTGGGCATCGTAAGTAT
	Nanog	Forward	GTCCCGGTCAAGAAACAGAA
		Reverse	TGCGTCACACCATTGCTATT
	OCT3/4	Forward	ATTCAGCCAAACGACCATCT
		Reverse	ACACTCGGACCACATCCTTC
	PAI-1	Forward	GACTCGTGAAGTCAGCCTGAAAC
		Reverse	GACTCGTGAAGTCAGCCTGAAAC
	ZEB1	Forward	GGCATACACCTACTCAACTACGG
		Reverse	TGGGCGGTGTAGAATCAGAGTC

qRT-PCR, quantitative reverse transcription PCR; 18S rRNA, 18S ribosomal RNA; GLUT3, glucose transporter 3; α-SMA, α-smooth muscle actin; CTGF, connective tissue growth factor; PAI-1, plasminogen activator inhibitor-1; ZEB1, zinc finger E-box binding homeobox 1.

## Data Availability

The data presented in this study are available on request from the corresponding author.

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
