# Peer review of "GLUT3 Promotes Epithelial–Mesenchymal Transition via TGF-β/JNK/ATF2 Signaling Pathway in Colorectal Cancer Cells"

_biomedicines, 2022, doi:10.3390/biomedicines10081837_

Round 1

Reviewer 1 Report

Dear authors,

You have done a great work, but there are some aspects that, to my point of view, need some changes or explanations.

1. To understand the increase of glucose metabolism in cancer cells, the Warburg effect should be explained.

2.  The sentence, between lines 37-39, “It is highly ex-37 pressed in various cancer cells including colon carcinoma, glioblastoma, hepatoblastoma and metabolic adaptation to nutrient deprivation in brain tumor-initiating cells.”, needs to be better explained.

3. In the results’ chapter, the first sentences of 3.1, 3.2, 3.3 and 3.4 would fit better in the introduction.

Author Response

Reviewer 1:

Dear authors,

You have done a great work, but there are some aspects that, to my point of view, need some changes or explanations.

Author reply: The authors acknowledge the reviewer’s favourble comments

1. To understand the increase of glucose metabolism in cancer cells, the Warburg effect should be explained.

Author reply: Thank you so much for your important comments. In accordance with the reviewer’s comments, we have added a description of the Warburg effect on lines 48-55 in Introduction part in the revised manuscript. Please check the revised manuscript for the modified Introduction section marked with highlight.

2. The sentence, between lines 37-39, “It is highly expressed in various cancer cells including colon carcinoma, glioblastoma, hepatoblastoma and metabolic adaptation to nutrient deprivation in brain tumor-initiating cells.”, needs to be better explained.

Author reply: In accordance with the reviewer’s comments, we modified and rewrote the 3rd paragraph in Introduction part on lines 58-64 in the revised manuscript. Please check the revised manuscript for the modified Introduction section marked with highlight.

3. In the results’ chapter, the first sentences of 3.1, 3.2, 3.3 and 3.4 would fit better in the introduction.

Author reply: In accordance with the reviewer’s comments, we deleted and/or modified the first sentences of 3.1, 3.2, 3.3 and 3.4 sections in Results part in the revised manuscript. Please check the revised manuscript for the modified Results and Introduction marked with highlight.

Reviewer 2 Report

It is a comprehensive and interesting study. It is necessary to carefully read and edit the English. Introduction, line 70 "molecular mechanisms correlation with GLUT3-induced EMT is not poorly understood"- then why you provided this research? Lines 201-203 from results section belong rather to discussion. Check the units "fold of control" in all figures fo accurance. Non-significance does not need to be highlighted (fig. 5). Lines 360-364 belong to discussion. 

Author Response

Reviewer 2:

It is a comprehensive and interesting study. It is necessary to carefully read and edit the English. 

Author reply: Thank you for the favourable comments. In accordance with the reviewer’s comments, the authors carefully read and corrected the English in the revised manuscript.

1. Introduction, line 70 "molecular mechanisms correlation with GLUT3-induced EMT is not poorly understood"- then why you provided this research?

Author reply: Thank you so much for your comments. In accordance with the reviewer’s comments, we modified and rewrote the 4th paragraph in Introduction part in the revised manuscript. The sentence has been modified to be better on lines 85-87 in the revised manuscript as follows.

2. Lines 201-203 from results section belong rather to discussion.

Author reply: In accordance with the reviewer’s suggestion, we deleted the first sentences of 3.1 sections in Results and moved the content to the first paragraph in Discussion in the revised manuscript. Almost all of the Discussion was rewritten. Please check the revised manuscript for the modified Results and Discussion section marked with highlight.

3. Check the units "fold of control" in all figures for accurance.

Author reply: In accordance with the reviewer’s comments, we have carefully checked the entire Figures. We modified the ‘Fold of control’ to ‘Relative mRNA level’ in Fig. 1B, Fig. 2B, Fig. 3B and 3D, and Fig. 5B in the revised manuscript.    

4. Non-significance does not need to be highlighted (fig. 5).

Author reply: In accordance with the reviewer’s comments, we deleted ‘n.s.’ and inserted statistical significance (P < 0.01) between the two groups of the GLUT3 promoter and the ATF-mutated GLUT3 promoter under TGF-beta treatment in Figure 5C in the revised manuscript.

5. Lines 360-364 belong to discussion.

Author reply: In accordance with the reviewer’s suggestion, we moved the content to the last paragraph in Discussion in the revised manuscript. Almost all of the Discussion was rewritten. Please check the revised manuscript for the modified Results and Discussion section marked with highlight.

Reviewer 3 Report

In this manuscript the Authors hypothesized that the expression of GLUT3 is associated with EMT progression in metastatic CRC, and that the molecular mechanisms supporting GLUT3-induced EMT involve TGF-β/JNK/ATF2 signaling pathway activation. The results showed that GLUT3 is regulated by ATF2 transcription factor through a SMAD-independent pathway mediated by JNK signaling pathway after TGF-β activation, leading to a GLUT3 overexpression which is associated with EMT genes expression and invasiveness.

Comments:

Which is the molecular mechanism that explain as GLUT3 overexpression is associated with EMT genes/proteins expression

Explain because for the EMT markers (N-cadherin, Snail, Twist, CTGF, fibronectin 1 and vimentin) was evaluated only the gene expression. Protein expression should be evaluated. Vice versa for PAI-1 and ZEB1

Lane 210 “the mRNA expression of mesenchymal genes including N-cadherin, ………. was increased”, in the figure 1B the N-cadherin is not overexpressed, please explain and correct

Protein expression levels of Bmi-1, Nanog and 264 OCT3/4 should be detected

Which is the expression of EMT markers when TGF-beta overexpression was induced??

Lanes 384-385 “However, comprehensive mechanism studies against GLUT3-induced EMT were still required”, it is not completely true because references 38 and 39 explain some mechanisms.

Lanes 401-404 “Indeed, our results showed that inhibition of TGF-β downregulates the phosphorylation of JNK, which results in increasing mesenchymal markers such as vimentin, fibronectin, α-SMA, whereas epithelial marker such as E-cadherin was decreased in CRC cells”. This phase is contradictory respect to the results.

The discussion section should be expanded, the manuscript does not include discussion on literature's wide data about this topic.

The conclusion should be rewritten because are redundant

Lane 305 update references and especially involving the role of JNK in the TGF-beta-induced EMT in human

Lane 306 update of references is necessary

Lane 314 eliminate (Figure 5 B)

Author Response

Reviewer 3:

In this manuscript the Authors hypothesized that the expression of GLUT3 is associated with EMT progression in metastatic CRC, and that the molecular mechanisms supporting GLUT3-induced EMT involve TGF-β/JNK/ATF2 signaling pathway activation. The results showed that GLUT3 is regulated by ATF2 transcription factor through a SMAD-independent pathway mediated by JNK signaling pathway after TGF-β activation, leading to a GLUT3 overexpression which is associated with EMT genes expression and invasiveness.

Comments

1. Which is the molecular mechanism that explain as GLUT3 overexpression is associated with EMT genes/proteins expression

Author reply: Jiang et al have reported that yes‑associated protein (YAP), as a major downstream effector in the Hippo signaling pathway, and AMPK pathway regulate the expression of GLUT3 and EMT process in human colorectal cancer tissues and colorectal cancer cells (Oncol. Lett., 2021). In addition, Kuo and collegues revealed that Hippo and YAP pathway is associated with promotion of CRC invasiveness and stemness by GLUT3 upregulation (Theranostics, 2019). Hippo-YAP pathway has been reported to be regulated by cellular energy status and is implicated in cancer metastasis and EMT process. Therefore, the Hippo-YAP pathway can be considered as one of the molecular mechanism explaining GLUT3 overexpression and EMT process. We described this mechanism in the first paragraph of the Discussion in the revised manuscript. Please check the revised manuscript for the modified Discussion marked with highlight.

2. Explain because for the EMT markers (N-cadherin, Snail, Twist, CTGF, fibronectin 1 and vimentin) was evaluated only the gene expression. Protein expression should be evaluated. Vice versa for PAI-1 and ZEB1.

Author reply: In accordance with the reviewer’s comments, we have conducted Western blot analysis to detect the protein expression of N-cadherin, Snail, Twist, CTGF, Fibronectin, Vimentin. We also performed qRT-PCR to measure the mRNA levels of PAI-1 and ZEB1. We have spent almost 7 days given to us to obtain antibodies, perform experiments and analyze the results. We changed the Figure 1-2 and also modified the Results section and Figure legends in the revised manuscript. Please check the revised manuscript for the modified Results section marked with highlight.

3. Lane 210 “the mRNA expression of mesenchymal genes including N-cadherin, ………. was increased”, in the figure 1B the N-cadherin is not overexpressed, please explain and correct

Author reply: In accordance with the reviewer’s comment, we modified and rewrote the sentences on line 223-230 in Results 3.1 section in the revised manuscript. Please check the revised manuscript for the modified Results section marked with highlight.

4. Protein expression levels of Bmi-1, Nanog and 264 OCT3/4 should be detected

Author reply: In accordance with the reviewer’s comments, we have conducted Western blot analysis to detect the protein expression of Nanog and OCT3/4. The protein expression of BMI-1 could not be measured due to lack of time and absence of antibody. We changed the Figure 3 and also modified the Results section and Figure legends in the revised manuscript. Please check the revised manuscript for the modified Results section marked with highlight.

5. Which is the expression of EMT markers when TGF-beta overexpression was induced??

Author reply: It is well known that TGF-β induces EMT process and plays an important role in cancer metastasis. Yu et al showed that activation of TGF-beta induced EMT process through decreased expression of F-actin and E-cadherin using transdifferentiation and fluorescence staining assay (EMBO J., 2002). Yeh et al showed that activation of TGF beta upregulates the expression of snail, slug, twist, nanog, oct4, sox2, etc in liver cancer cells (Nature Cell Biol., 2018). We described these contents in the 3rd paragraph of the Discussion in the revised manuscript. In this study, we showed that GLUT3 induced the expression of EMT markers and the treatment with TGF beta induced the expression of GLUT3 in CRC cells. These results are in agreement with the previous studies demonstrating that TGF-β pathway upregulates the EMT process in cancer. Therefore, we did not experimentally show the expression changes of EMT markers caused by TGF beta overexpression. Please check the revised manuscript for the modified Discussion marked with highlight.

6. Lanes 384-385 “However, comprehensive mechanism studies against GLUT3-induced EMT were still required”, it is not completely true because references 38 and 39 explain some mechanisms.

Author reply: In accordance with the reviewer’s suggestion, we have added some mechanisms for GLUT3-induced EMT and rewrote the first paragraph in Discussion in the revised manuscript. Almost all of the Discussion was rewritten. Please check the revised manuscript for the modified Discussion section marked with highlight.

7. Lanes 401-404 “Indeed, our results showed that inhibition of TGF-β downregulates the phosphorylation of JNK, which results in increasing mesenchymal markers such as vimentin, fibronectin, α-SMA, whereas epithelial marker such as E-cadherin was decreased in CRC cells”. This phase is contradictory respect to the results.

Author reply: In accordance with the reviewer’s comment, we modified and rewrote the Discussion of the revised manuscript. Most of the Discussion part was rewritten. The sentence pointed out by the reviewer was written on line 462-467 in 4th paragraph of the Discussion in the revised manuscript. Please check the revised manuscript for the modified Discussion section marked with highlight.

8. The discussion section should be expanded, the manuscript does not include discussion on literature's wide data about this topic.

Author reply: In accordance with the reviewer’s comment, we rewrote the Discussion of the revised manuscript. Almost all of the Discussion was rewritten. Please check the revised manuscript for the modified Discussion section marked with highlight.

9. The conclusion should be rewritten because are redundant

Author reply: In accordance with the reviewer’s comment, we rewrote the Conclusion of the revised manuscript. Please check the revised manuscript for the modified Conclusions marked with highlight.

10. Lane 305 update references and especially involving the role of JNK in the TGF-beta-induced EMT in human

Author reply: We modified and rewrote the Results and Discussion in the revised manuscript. The sentence pointed out by the reviewer was written in 4th paragraph of the Discussion in the revised manuscript. In accordance with the reviewer’s suggestion, we updated the reference on line 460-462 (Ref. 64-68) of the revised manuscript. Please check the revised manuscript for the modified Results section marked with highlight.

11. Lane 306 update of references is necessary

Author reply: We modified and rewrote the Results and Discussion in the revised manuscript. The part pointed out by the reviewer was written in 5th paragraph of the Discussion in the revised manuscript. In accordance with the reviewer’s suggestion, we updated the reference on line 470-472 (Ref. 68-69) of the revised manuscript. Please check the revised manuscript for the modified Results section marked with highlight.

12. Lane 314 eliminate (Figure 5 B)

Author reply: In accordance with the reviewer’s comment, we modified and rewrote the Results 3.4 section in the revised manuscript. Most of the Results 3.4 was rewritten. Please check the revised manuscript for the modified Results 3.4 section marked with highlight.

Round 2

Reviewer 3 Report

None